# Measuring Transverse Displacements Using Unmanned Aerial Systems Laser Doppler Vibrometer (UAS-LDV): Development and Field Validation

**DOI:** 10.3390/s20216051

**Published:** 2020-10-24

**Authors:** Piyush Garg, Roya Nasimi, Ali Ozdagli, Su Zhang, David Dennis Lee Mascarenas, Mahmoud Reda Taha, Fernando Moreu

**Affiliations:** 1Department of Electrical & Computer Engineering, University of New Mexico, Albuquerque, NM 87131-0001, USA; pgarg@unm.edu; 2Department of Civil, Construction, & Environmental Engineering, University of New Mexico, Albuquerque, NM 87131-0001, USA; rhnasimi@unm.edu (R.N.); aozdagli@unm.edu (A.O.); suzhang@unm.edu (S.Z.); mrtaha@unm.edu (M.R.T.); 3Earth Data Analysis Center (EDAC), University of New Mexico, MSC01 1110, 1 University of New Mexico, Albuquerque, NM 87131-0001, USA; 4Los Alamos National Laboratory, National Security Education Center (NSEC), Los Alamos National Labs MS-T001, PO Box 1663, Los Alamos, NM 87545, USA; dmascarenas@lanl.gov

**Keywords:** unmanned aerial system, reference-free displacement, non-contact displacement, laser, railroad bridge, field implementation, dynamic displacement

## Abstract

Measurement of bridge displacements is important for ensuring the safe operation of railway bridges. Traditionally, contact sensors such as Linear Variable Displacement Transducers (LVDT) and accelerometers have been used to measure the displacement of the railway bridges. However, these sensors need significant effort in installation and maintenance. Therefore, railroad management agencies are interested in new means to measure bridge displacements. This research focuses on mounting Laser Doppler Vibrometer (LDV) on an Unmanned Aerial System (UAS) to enable contact-free transverse dynamic displacement of railroad bridges. Researchers conducted three field tests by flying the Unmanned Aerial Systems Laser Doppler Vibrometer (UAS-LDV) 1.5 m away from the ground and measured the displacement of a moving target at various distances. The accuracy of the UAS-LDV measurements was compared to the Linear Variable Differential Transducer (LVDT) measurements. The results of the three field tests showed that the proposed system could measure non-contact, reference-free dynamic displacement with an average peak and root mean square (RMS) error for the three experiments of 10% and 8% compared to LVDT, respectively. Such errors are acceptable for field measurements in railroads, as the interest prior to bridge monitoring implementation of a new approach is to demonstrate similar success for different flights, as reported in the three results. This study also identified barriers for industrial adoption of this technology and proposed operational development practices for both technical and cost-effective implementation.

## 1. Introduction

The U.S. railroad network transports up to 40% of the total cross-country freight [1,2]. There are approximately 100,000 railroad bridges [3] on this network, which is about 225,000 km long [4], making the performance of the bridges very critical for the safe operation of the rail networks. Maintenance of bridges is of topmost priority for railroad bridge engineers [5] as about 50% of the North American railroad bridges are more than 100 years old [6]. Railroad managers maintain bridges by inspecting them on a regular basis to ensure their safety and to prevent derailments, delay in network operation, and loss of time and resources. Most of the current methods of bridge inspection require visual inspection [7], and oftentimes visual inspection is not reliable [8]. Structural health monitoring (SHM) of a bridge can assist in determining its operational capabilities. Railroad management agencies are interested in objectively informing the safety and growth of their operations with measurement of parameters and responses under loads. Bridge displacement measurement under dynamic loading due to the train movement is an important aspect of SHM [9]. According to past research related to bridge safety and performance, the changes of transverse dynamic displacements amplitudes under train crossing events across time can be an indication of change of bridge condition. More specifically, railroads are interested to quantify the changes of bridge responses to forced vibrations caused by trains over time. However, there are limitations to collect transverse dynamic displacement time histories in outdoor environments.

Contact sensors such as Linear Variable Differential Transducer (LVDT) [10] and accelerometers [11,12,13] are traditionally used for measuring transverse bridge displacement under train loading. Appropriately mounting an LVDT in the field is challenging since they require a fixed reference from where to measure. As the majority of bridges are across the water and in remote locations, constructing the reference is often difficult [10]. Alternatively, researchers have used reference-free, contact sensors as accelerometers to estimate displacements. However, their output is not always reliable as they require double integration to obtain the displacement data, which generally adds a drift [13]. Global Positioning Systems (GPS) have also been used as contact sensors for displacement measurement [14,15,16]. However, these systems are not always feasible as the units’ accuracy is not adequate for detecting small displacements generated by dynamic train loading.

To address the issues of contact sensors, non-contact sensors have been researched for bridge displacement measurement. The robotic total station (RTS), which autonomously identifies and tracks the target, has been researched as a tool for non-contact displacement measurement, but the success of the sensor depends on suitable atmospheric conditions [17]. Image processing has also been explored extensively for non-contact displacement measurement using commercial cameras [18,19,20,21]. This method requires extensive postprocessing and relatively complicated algorithms to obtain an accurate output. The equipment also needs to be set up close to the target, and the light conditions need to be ideal for image and video capturing. These issues indicate that the use of image processing is not always feasible for railroad bridge displacement measurements in field implementations.

Unmanned aircraft system (UAS) platforms have been used in several different venues, such as inspection, aerial photography, surveillance, and remote sensing [22]. The use of UAS offers more flexibility and easier access to structures, which was previously not possible [23]. Due to their agility, UAS have found their use in several applications such as reconnaissance and disaster management [24], oil spill surveillance and detection [25], soil erosion monitoring [26], forest ecosystem and biodiversity monitoring [27], and deforestation detection [28]. Researchers have used UAS-based systems for bridge SHM in recent years [29,30,31,32,33]. Novel applications of UAS included non-destructive testing of concrete by tap testing it with a hammer mounted UAS [34]. There have been several attempts to use a camera-mounted UAS for bridge SHM [35,36]. 3D image correlation on aerial images captured by cameras on the UAS [37] and close-range photogrammetry or structure-from-motion (SfM) using a UAS [38,39,40] has also been investigated for SHM. Using cameras and other devices mounted on UAS for SHM holds the potential to solve the problems related to accessibility in remote locations and hazardous conditions. These methods still require a reference for image processing, postprocessing of the captured data, and algorithms to extract valuable information from the collected data. Additionally, cameras are unable to obtain out-of-plane displacements, or transverse displacements. Thus, such methods fail to address the major shortcomings of the traditional methods of SHM.

The use of aerial photography in combination with GPS to detect thin cracks in concrete surfaces has also been demonstrated [41]. Light detection and ranging (LiDAR) is a laser light-based sensor that produces a point cloud that can assist users in developing and analyzing the 3D model of the infrastructure [42]. The UAS-based LiDAR system has been examined as a sensor for infrastructure management [43]. Airborne LiDAR systems have also been used in modeling the buildings and detecting damages [44] as well as in topographical mapping features [45]. The airborne LiDAR systems have been also used in the detection of dynamic activities, such as landslide mapping and damage assessment [46], building and infrastructure change detection over time [47], and volumetric changes in coastal dunes detection [48], and displacement [49]. Laser Doppler Vibrometer (LDV) measures target vibration using frequency change in the transmitted laser signal due to the Doppler effect. LDV as a non-contact sensor has been used for bridge displacement measurement [50,51]. While this device still needs to be placed on a rigid surface near the target, the operation distance is usually long and is hindered less by the terrain conditions. Additionally, the output of LDV is less dependent on visibility and atmospheric conditions and is real-time, requiring minimal postprocessing. Although these advantages make the use of LDV ideal for dynamic bridge displacement measurement, it still not reference-free. The authors developed in the past one system to overcome the challenge of noncontact displacement measurement using a moving LDV [50]. The results of this paper showed laboratory estimations and an outdoor flight, lacking repeatability in outdoor environments, which according to railroad managers it is required prior to railroad bridge monitoring under real trains. According to the owners, the measurement of transverse displacements with a moving LDV on an UAS needs to be tested at different conditions, as laser data depends on the range and the UAS flight and hovering changes for each event. It would be of value to the railroad industry to conduct multiple tests that can provide evidence through field validation and development. Outdoor capabilities of this new system require testing the method at different distances and different displacements, and to identify barriers for implementation. According to the railroads, it would be of value to design the algorithm with hovering properties to ensure the method is repeatable for each flight in the future.

This study proposes the mounting of an LDV on a UAS for measuring dynamic displacement under train loading, validating, and developing a system through a series of field experiments of different distances and displacements. The LDV was mounted onto the UAS, and the assembly was tethered to the ground for safety and, at the same time, to protect the equipment from any major damage. A wooden plank was moved to simulate the movement of a bridge and was used as a target for the LDV to track. The readings from the LDV were compared with the output of an LVDT attached to the plank to demonstrate the effectiveness of using an Unmanned Aerial Systems Laser Doppler Vibrometer (UAS-LDV) for bridge displacement measurement. The authors conducted a comparison between the measured dynamic displacement using the proposed method and the measured dynamic displacement by an LVDT for validation. Finally, barriers of using LDV on an UAS for field applications were identified, and possible solutions for each of the issues were suggested.

## 2. Equipment Selection

This section discusses the displacement measurement of a vibrating target using a Laser Doppler vibrometer mounted on a UAS. This section begins with introducing the LDV and UAS used for the research. Subsequently, the hovering motion of the UAS is measured, and a filter is designed to compensate for this UAS movement. Finally, researchers conducted a series of outdoor experiments to validate the ability of the new system to measure dynamic displacements comparing the LDV measurements from the UAS with data collected using one LVDT that was brought to the field to serve as a fixed reference (ground truth). Both signals were filtered to validate the ability of the UAS-LDV system to measure non-contact, reference-free dynamic displacements. The vibrometer used for this research is Polytec OFV 534. The vibrometer controller compatible with this vibrometer is Polytec OFV5000. The sensitivity of the controller module is set to 5 mm/V. The data acquisition system used for these experiments is m+p international made VibPilot, which has a range of ± 10 V and can sample data at 204.8 kHz with 24-bit quantization. At the times when the recorded displacement is more than ± 50 mm, the voltage signal saturates. The “return-to-zero” feature built in the vibrometer controller shifts the voltage signal to the zero level when the signal goes out of measurement range without interrupting the recording. The sensor head and LDV module weigh 1.0 kg and 4.2 kg, respectively, [52] excluding the controller. To be considered for the research, the UAS needs to be capable of hovering with a payload of more than 5.2 kg. Moreover, the system needs to be designed and validated to eventually be used for testing of railroad bridges under dynamic loading. The average length of a conventional freight train is about 2 km [53], and the average freight speed is approximately 30–35 km/h (for both tonnage and manifest trains) [54]. A freight can last three to four minutes to cross a typical railroad bridge of 150 m long. Thus, to observe and record complete train crossing events, the UAS-LDV assembly should be able to hover for some time before and after the train. Researchers discussed various UAS systems with railroad owners in order to (1) ensure the UAS is able to carry the LDV weight of 5.2 kg (1 kg and 4.2 kg for sensor head and laser unit, respectively) and (2) propose a cost-effective first prototype that can be affordably repaired or replaced in the event of crash during experimentation and can hover for more than 15 min for displacement monitoring in the field. Researchers selected DJI Matrice 600 Pro for this research (Figure 1). The UAS has an approximate hovering time of 18 min at a full payload of 5.5 kg [55]. It also has an enhanced GPS module and an inertial measurement unit (IMU), along with self-correction and stabilization capabilities. All these features make this UAS suitable for this research and its outdoor implementation.

Researchers used the LDV to measure the hovering motion of one UAS under normal field conditions. Figure 2 illustrates the field test setup where Polytec LDV records the movement of a DJI Phantom 3 Pro. Although the two UAS differ in both hardware and software, this preliminary exploration was conducted for preparation purposes and only as exploratory research, given that there was no information available about the displacement of UAS using a fixed reference.

To understand the hovering motion of the UAS, the time domain data is converted into the frequency domain. Figure 3 shows the discrete Fourier transformation analysis of the UAS hovering signal into the frequency domain. It is observed that most of the power in the hovering is concentrated around the frequencies under 0.5 Hz. For the subsequent sections, the LDV signal will be filtered using a high-pass filter to eliminate the low-frequency hovering motion of the UAS and to compare the dynamic motions of vibrometer and LVDT. Moreu et al. [10] reported that the measurement of the high-frequency dynamic (not low-frequency pseudo-static) transverse displacements of railroad bridges can inform railroad managers of the condition of the bridges. The research literature identifies the pseudo-static frequency of railroad timber bridges at 0.5 Hz [56]. Researchers found from the three experiments that the total UAS-LDV measurements included both transverse board displacement and the UAS movement, both of low frequency. Railroad owners are interested to measure dynamic time history of displacements to quantify safety. The low frequency components of the laser measurement caused by both the UAS hovering and the board movement were filtered with a high-pass filter. Using the three experiments, the research team selected a Butterworth filter of order three, with a −3 dB (half-power) or a cut-off frequency of 0.5 Hz that removes the hovering of UAS and pseudo-static movement of the board. Figure 4 summarizes the designed filter. Similarly, the LVDT data of the board movement were filtered to get the dynamic displacement with the same filter.

## 3. Field Testing

In this section, the experimental layout for displacement measurement using an LDV mounted on UAS, and its implementation for field testing are explained in detail. Figure 5 shows the experimental layout for field testing using the LDV mounted onto the UAS. The measurements obtained by the vibrometer are compared to the actual displacement of the target recorded by LVDT. The connection between the vibrometer and its data acquisition unit is a fixed optical fiber cable. Thus, to protect the vibrometer and to prevent injuries and damage in the event of sudden and unexpected drone movement, the UAS is tethered to the ground using a heavyweight.

The UAS is made to hover about 1.5 m above from the ground. The UAS is flown at 4 to 7 m away from the target, and the LVDT is arranged to measure the same location on the target where UAS is pointing. The authors have used the data from ten train crossing events over a real timber railroad bridge to conduct their experiment with realistic movements [5,10]. Figure 6 displays the implementation of the field test set-up. The plank is manually moved in such a way that it simulates the movement of the railway bridge with multiple frequencies and amplitude components, including the pseudo-static displacement (Figure 6a). The validation of the UAS-LDV is critical for the field implementation, and it is ensured by measuring one side of the target and collecting LVDT data on the opposite side of the target (Figure 6b,c), respectively).

## 4. Results and Discussion

The goal of this research was twofold: (1) to develop a new implementation for UAS and LDV that can successfully measure dynamic displacements and (2) to validate its accuracy and repeatability. The research team designed and tested the new system prior to the displacement validation to ensure the UAS could carry the weight of the LDV. The added weight caused the UAS to crash in two occasions, given the added eccentricity. The research team used the lessons learned from the unsuccessful field trials and (1) tested different, incremental weight increased in outdoor test flights, and resolved the issue only carrying the sensor head and tethered the laser unit to the ground; (2) tested different weather conditions, elevations, and speeds of UAS prior to the dynamic measurement; and (3) incrementally training the pilot to test the ability to hovering the UAS near a surface, prior to LDV sensing. The results were (1) increased control of UAS hovering; (2) incremental changes and modifications in the design of a UAS system which is critical for flight control, and improved design that is compatible with flying the LDV with the UAS safely; and (3) preliminary testing prior to dynamic experiments that ensured the hovering of the UAS and the testing procedures were safely understood by the pilot, which is critical for the outdoor testing protocol development.

Of the three successful trials, one was captured from a distance of 4 m from the target, and two from a distance of 7 m from the target. Figure 7 shows the raw data captured by the vibrometer data acquisition system for these three trials. During the three experiments, the LDV exceeds the range during the hovering. The LDV data exceeds the range due to the UAS hovering in excessive transverse or lateral directions with respect to the target, and the measurement “returns-to-zero” as explained in Section 2. The range can be exceeded either if the UAS moves away from the target beyond the range in the transverse direction LDV, or by the UAS rotating or translating laterally, which will cause the LDV reading to fall outside the target. During the various field validation tests, the following conclusions can be made. (1) In general, the pilot skills were not sufficient to control the inherent hovering during testing, so LDV readings in general fell out of range at certain periods of time during the monitoring; (2) out of range events were automatically corrected with a “return-to-zero” feature built in the vibrometer controller, with close to zero disruption of the signal; and (3) as a result, the displacement accuracy can only be trusted in the regions in between “out of range” events. Figure 8 shows the data captured by the vibrometer data acquisition system for these three trials corrected with “return-to-zero” feature. Figure 8 shows that the output is drifting due to the motion of the UAS at low frequency.

The frequency-domain analysis of the signals from the vibrometer and the LVDT is shown in Figure 9 at different resolution levels and frequency ranges. Figure 9a shows that the motion of the UAS adds spurious components to the response, especially at the lower frequencies. Figure 9b shows that the signals from the vibrometer as well as the LVDT show a similar profile at frequencies greater than 0.5 Hz. Both signals from LDV and LVDT are filtered using a high-pass filter with a cut-off frequency of 0.5 Hz as explained in Section 2, which is acceptable for timber railroad bridges. The plot of these signals is shown in Figure 10. It is observed that the two signals match for the general dynamic displacement in the time domain. Figure 11 shows the filtered signals enlarged to the selected regions where the measurement was within range and when the LDV measurement was in target. For the selected regions with stable UAS and continuous LDV monitoring, the LDV and LVDT dynamic signals match both in amplitude and phase. The next section calculates and quantifies the errors.

The readings of the vibrometer in the UAS are compared to the measurements from the LVDT to benchmark the operational capabilities of the vibrometer. As the comparison is made between the measurements from two different sensors, the difference in measurement is not treated as percentage error but just as a percentage difference. For these experiments, two performance metrics are introduced: (i) the Max difference (*E*_1_) and (ii) RMS difference (*E*_2_).

The maximum difference between the signals is obtained by comparing the values at each of the sampling points. For “*n*” sampling points, the difference is obtained as
(1)$$E_{1}\left(i\right)=\left(abs\left(LVDT\left(i\right)-LDV\left(i\right)\right)\right),\;1\leq i\leq n$$

Thus, the maximum percentage difference can be obtained from Equation (2) as
(2)$$E_{1}\left(\%\right)=\left(\frac{max\left(E_{1}\right)}{max\left(abs\left(LVDT\right)\right)}\right)*100$$

The RMS difference for “*n*” sampling points is obtained as
(3)$$RMDS=\sqrt{\frac{\sum_{i=1}^{n}{\left(LVDT\left(i\right)-LDV\left(i\right)\right)}^{2}}{n}}$$

Thus, by using Equation (3), the percentage RMS difference normalized by range is
(4)$$E_{2}\left(\%\right)=\left(\frac{RMSD}{\max\left(LDV\right)-\min\left(LDV\right)}\right)*100$$

Both peak and RMS difference is less than 2 mm (Figure 12). Figure 13 shows that the average output peak error of the three tests is about 10%, and the average RMS difference is approximately 8%. These results indicate that for the three experiments the LDV mounted on a UAS was able to monitor the dynamic transverse displacements of a mocked timber railroad bridge with an approximate error of 10%, with peak displacements under 2 mm for bridge displacements in the order of magnitude of 20 mm (peak to peak). These levels of accuracy are acceptable for field decisions concerning timber railroad bridge performance. The interest in implementation prior to bridge monitoring of the new approach is to demonstrate similar success for different flights, as reported in the three results. The average value supports the capability of the three experiments, and it is worth mentioning that the railroad requires repeatability evidence prior to testing this approach in real bridges. The results reported herein support future field implementation.

## 5. Potential and Barriers for Field Implementation

The railroad industry has identified that this study can be suitable for timber railroad bridges implementation after resolving various barriers learned from the three experiments. According to Class I railroad bridge managers in North America, the most important contribution of a new methodology is its development and outdoor validation prior to the industrial application in the field with trains. Real field implementation can only be conducted after comprehensive outdoor testing, identifying barriers, and sharing these barriers and lessons learnt with the community. This section discusses the feasibility of the proposed method, potential barriers introduced by hardware integration, and recommends solutions for an integration in the field by industry. The first added requirement for field implementation is the development of an untethered solution once the system has been demonstrated to be satisfactory with the proposed hardware integration. To this end, the future development towards field implementation requires decreasing the payload added to the drone, including lighter LDV module, DAQ system, and power generator (or battery). Railroad bridge managers and inspectors prioritize the development of this solution with cost-efficient components that enables their adoption by industry, with an emphasis on small and mid-size railroads, which can benefit from using this approach to collect useful displacement information to prioritize repairs within their bridge management program. Table 1 identifies three barriers for field adoption for railroad bridge displacement monitoring, the specific hardware components that are associated to each impediment for adoption, and the proposed solutions that were identified to overcome these barriers. The first two barriers for adoption are interrelated, as well as the proposed solutions. Based on the results of this research, the authors recommend adopting a lighter, low-cost LDV system, with a shorter range. There are alternative distance lasers with shorter range such as low-cost laser sensors that can collect submillimeter accuracy from 1 or 2 m [57]. These sensors could be powered at low voltage so the tether to the power supply can be removed and the entire system can be carried by the UAS. A new low-cost DAQ system can be developed using wireless modules to record data remotely so there is no need for tethering the data to the ground.

## 6. Conclusions

Railroad managers and bridge inspectors need to interrupt traffic to install sensors in bridges, which has proven to be challenging and ineffective. This research describes the design of an integrated UAS and LDV system for field implementation to measure contact-free, reference-free displacements of railroad bridges under trains loads. Field tests were designed to develop a new system and validate its accuracy, by simulating the displacement of a railroad bridge under dynamic train loading. One preliminary result included two crashes of the new system given the lack of eccentricity of the LVD and the heavy weight of the new system. Researchers then evolved the new prototype incrementally to ensure safe operations prior to the testing for measurements in the field. The main outcomes include (i) the practical selection of a UAS suitable for lifting the vibrometer and able to hover near the structure being monitored, (ii) testing the mounting of LDV on a UAS for collection of vibration data and design and development of field testing for data collection, and (iii) successful testing and analysis of various field data collected by the LDV to demonstrate that this technique is feasible for bridge displacement monitoring. The results were analyzed in time and frequency domain. The analysis revealed that low-frequency components of the vibrometer readings included the movement of the UAS. After filtering both the LDV and LVDT signals for this low-frequency component, the responses from both sensors were compared to estimate dynamic bridge displacements that are non-contact and reference-free. The comparisons show that the LDV data collected with the UAS matched the LVDT measurements closely for all the field experiments. The average peak difference between the actual displacement captured by the LVDT and the displacement captured by the LDV is about 10%, and the average RMS difference is around 8%. It is evident that a laser-aided UAS can be considered as a viable alternative for displacement monitoring railroad bridges and offers minimal intrusion to the railroad traffic operations. A low-cost, untethered development of a new system that can collect total displacements under train crossing events is suggested. Based on the successful displacement collection obtained with the field experiments, the development of the untethered system has been identified as the top priority towards field implementation.

## Figures and Tables

**Figure 1 sensors-20-06051-f001:**
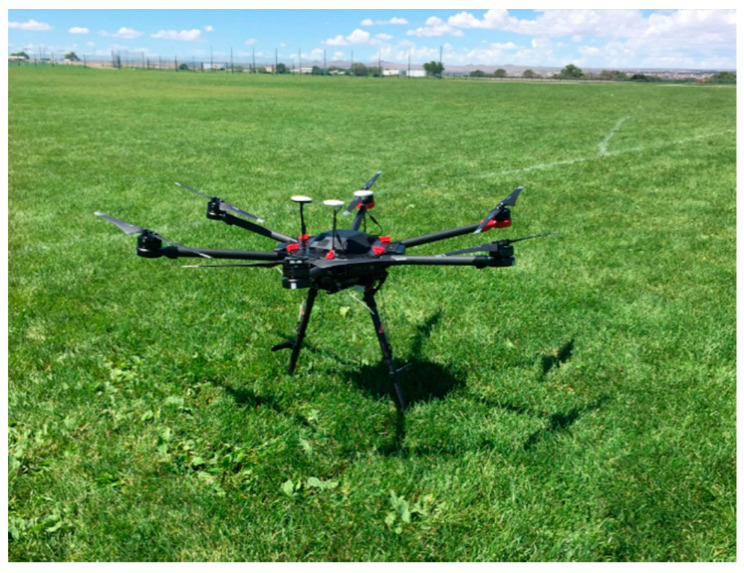
Unmanned Aerial System: DJI Matrice 600 Pro.

**Figure 2 sensors-20-06051-f002:**
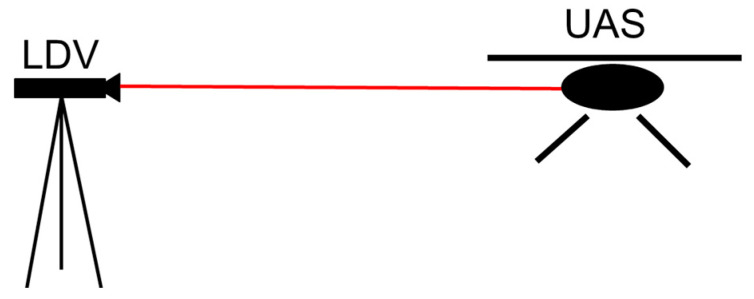
Field test setup to measure Unmanned Aerial System (UAS) hovering data.

**Figure 3 sensors-20-06051-f003:**
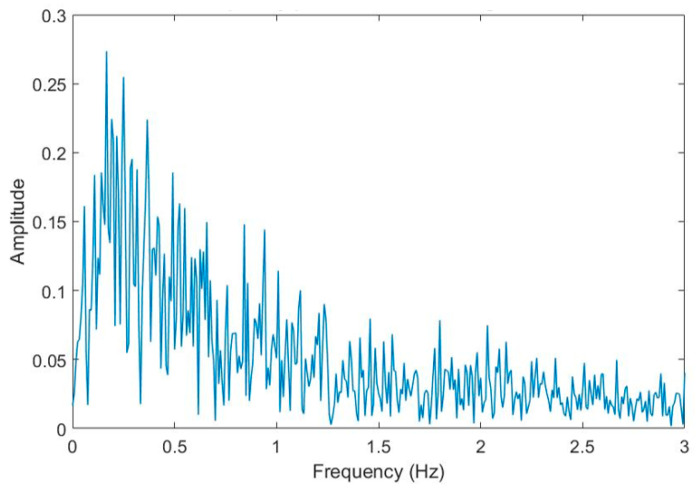
The frequency spectrum of UAS hovering motion.

**Figure 4 sensors-20-06051-f004:**
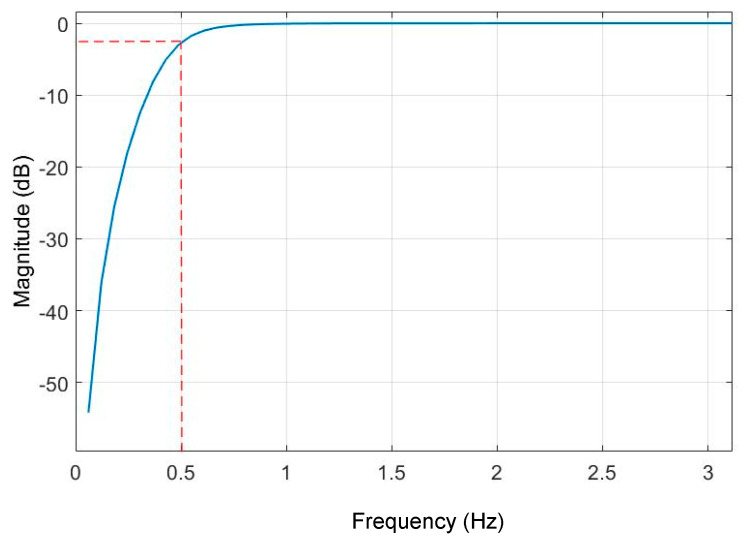
The magnitude response of 3rd-order Butterworth high-pass filter with a cut-off frequency of 0.5 Hz.

**Figure 5 sensors-20-06051-f005:**
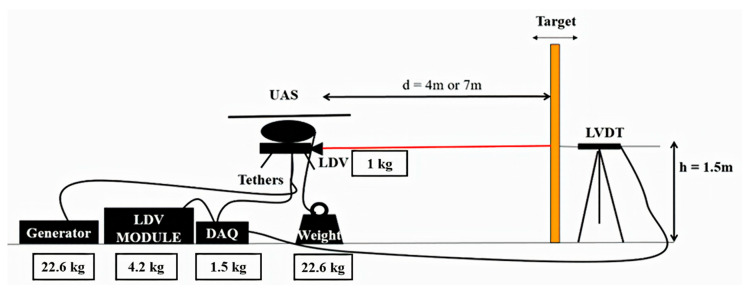
Experimental layout for field testing using Laser Doppler Vibrometer (LDV) mounted on an UAS.

**Figure 6 sensors-20-06051-f006:**
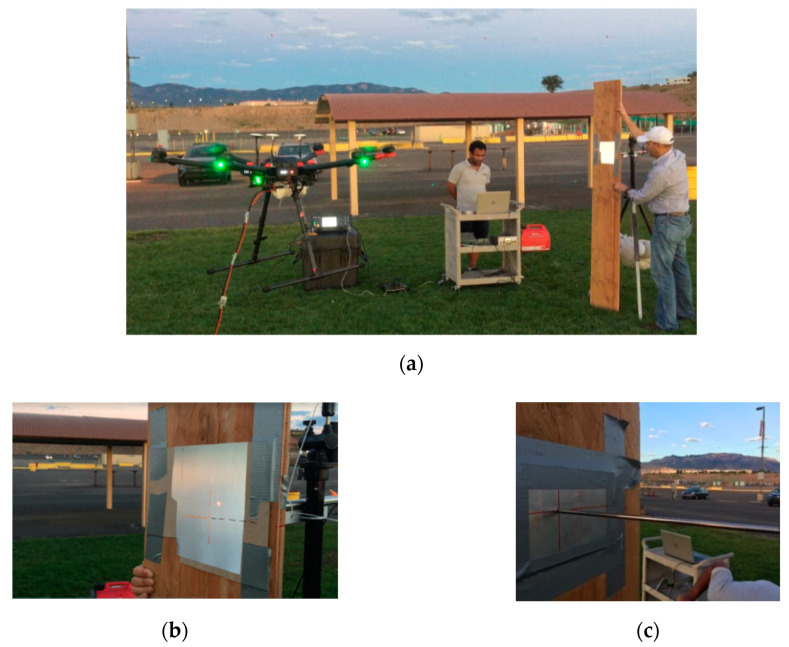
Outdoor validation: (**a**) bridge movement using a board with timber bridge displacement data, (**b**) LDV point in front of the board, and (**c**) LVDT validation at the opposite side.

**Figure 7 sensors-20-06051-f007:**
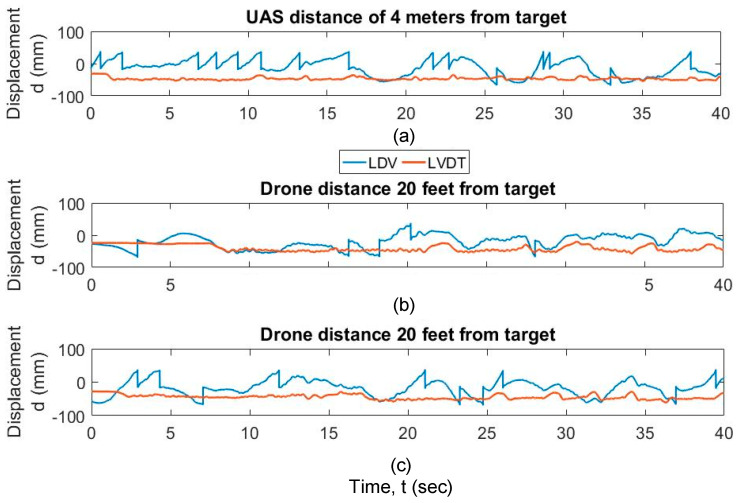
Raw LDV and LVDT output data for (**a**) LDV distance of 4 m from the target; (**b**) LDV distance of 7 m from the target (trial 1); and (**c**) LDV distance of 7 m from the target (trial 2).

**Figure 8 sensors-20-06051-f008:**
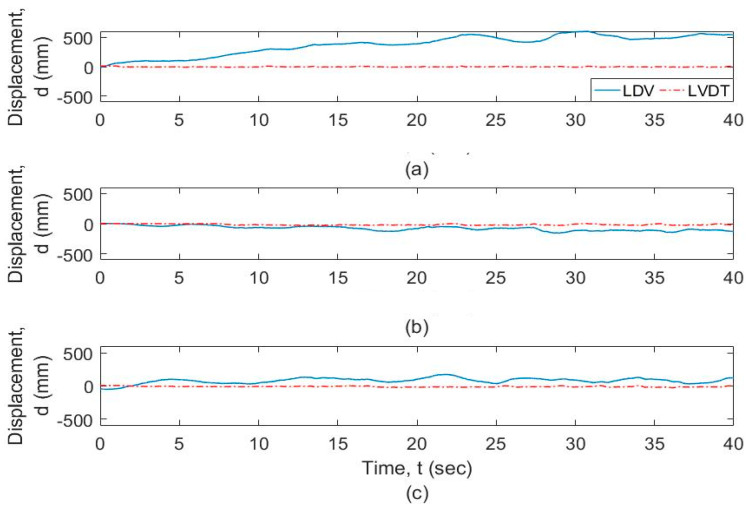
Comparison of LDV and Linear Variable Displacement Transducers (LVDT) output for (**a**) LDV distance of 4 m from the target, (**b**) LDV distance of 7 m from the target (trial 1), and (**c**) LDV distance of 7 m from the target (trial 2).

**Figure 9 sensors-20-06051-f009:**
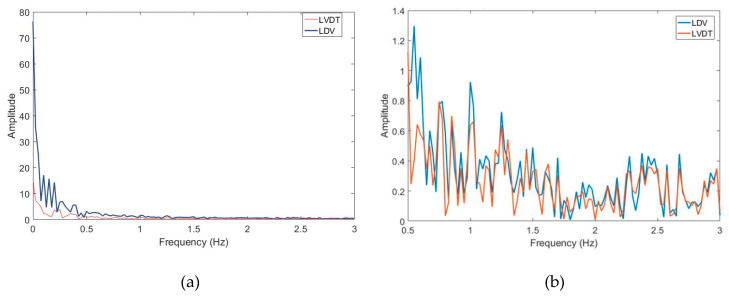
Comparison of signals of the LDV mounted on the UAS and LVDT when sensing at 7 m from the target: (**a**) total comparison (0–3 Hz) and (**b**) zoomed comparison (0.5–3 Hz).

**Figure 10 sensors-20-06051-f010:**
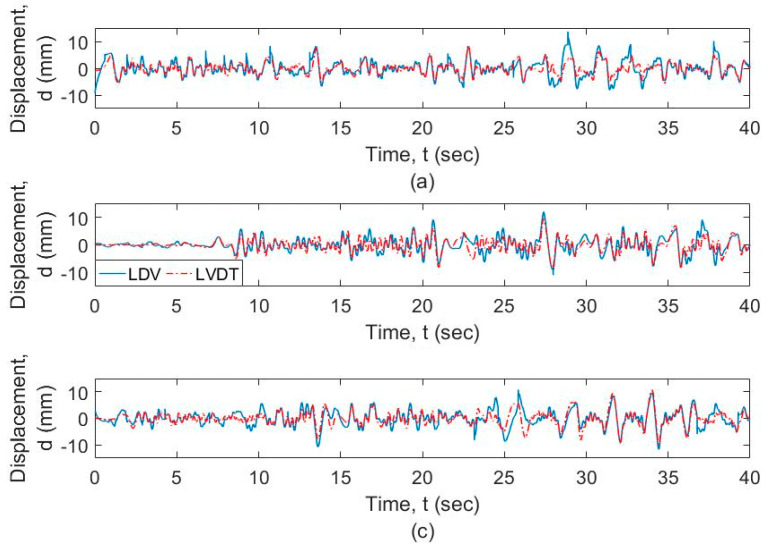
Filtered signal from LDV mounted on the UAS versus filtered LVDT signal for (**a**) LDV at a distance of 4 m from the target, (**b**) LDV at a distance of 7 m from the target (trial 1), and (**c**) LDV at a distance of 7 m from the target (trial 2).

**Figure 11 sensors-20-06051-f011:**
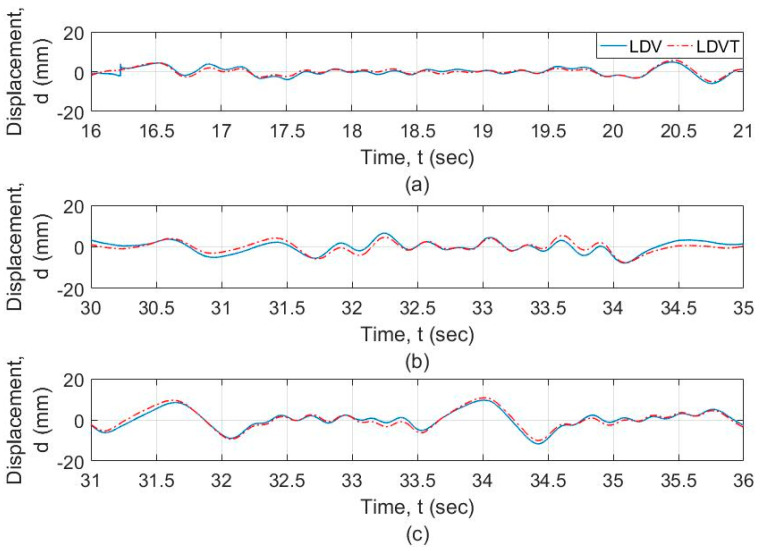
Filtered and focused signal from LDV mounted on the UAS versus LVDT signal for (**a**) LDV at a distance of 4 m from the target, (**b**) LDV at a distance of 7 m from the target (trial 1), and (**c**) LDV at a distance of 7 m from the target (trial 2).

**Figure 12 sensors-20-06051-f012:**
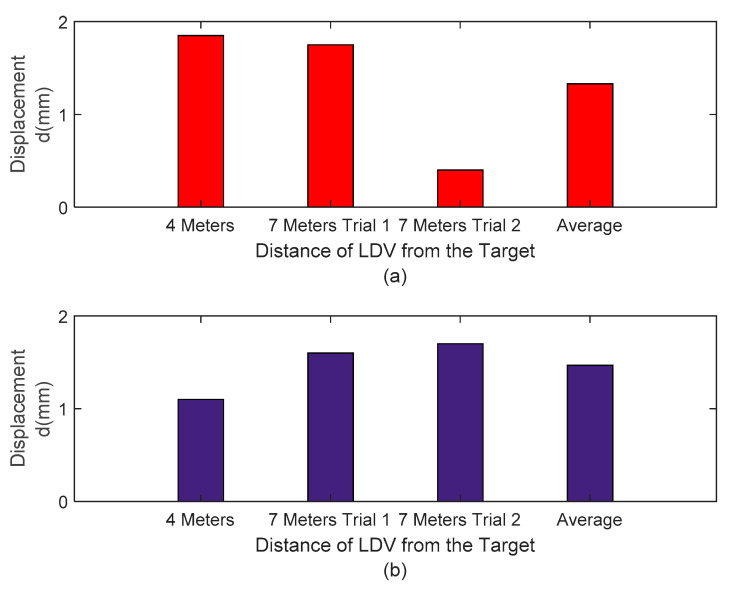
(**a**) Peak signal difference comparison between filtered LVDT and LVD signals and (**b**) RMS signal difference comparison between filtered LVDT and LDV signals.

**Figure 13 sensors-20-06051-f013:**
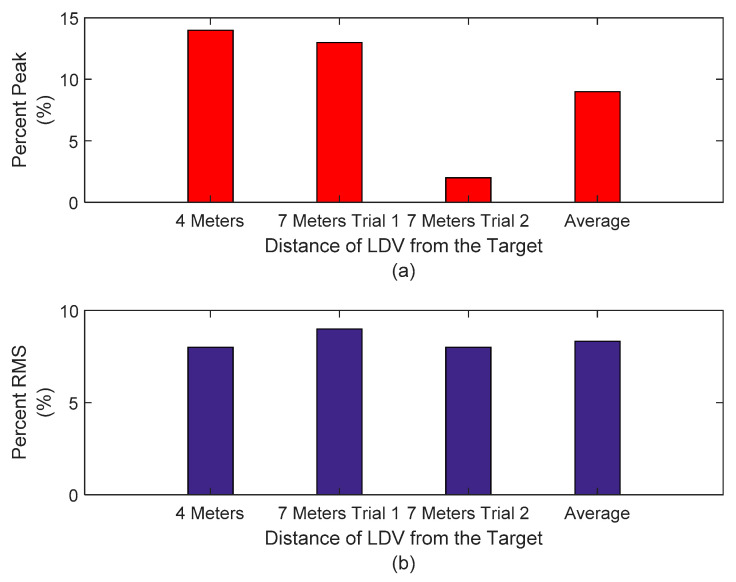
(**a**) Peak signal difference comparison between filtered LVDT and LDV signals in percentage and (**b**) RMS signal difference comparison between filtered LVDT and LDV signals in percentage.

**Table 1 sensors-20-06051-t001:** Hardware barriers and proposed solutions to enable field adoption of Unmanned Aerial Systems Laser Doppler Vibrometer (UAS-LDV) systems for railroad bridge displacement monitoring.

Barrier	Component	Proposed Solution
Expensive	LDV	For UAS applications, the UAS does not need to be hundreds of meters away from the moving object. Explore alternative lasers with fewer requirements.
Heavy	LDV	For lasers with a shorter range, the weight drops considerably. Explore lasers with shorter ranges (1 or 2 m, instead of hundreds of meters)
Tethered	UAS	Study integration of a wireless laser in the UAS, by designing and developing a light DAQ

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
