# Peer review of "Measuring Transverse Displacements Using Unmanned Aerial Systems Laser Doppler Vibrometer (UAS-LDV): Development and Field Validation"

_sensors, 2020, doi:10.3390/s20216051_

Round 1

Reviewer 1 Report

This paper proposes a reference-free displacement monitoring technique using a LDV that mounted on a UAS. The title mentioned field implementation, whereas there is just a test on a timber plank and no validation on real bridges. The applicability and effectiveness of the proposed technique are not well investigated. This paper is recommended to be major revised. The following points are suggested to be considered.

  1. The proposed method, especially the methodology, has been published in your previous work “Noncontact Dynamic Displacement Measurement of Structures Using a Moving Laser Doppler Vibrometer, J. Bridge Eng., 2019, 24(9): 04019089”. Please clearly state the difference/improvement of the present paper.
  2. Page 3, line 97, “LiDAR is not fast enough to detect real-time changes”. In practice, the LiDAR has a sampling rate of 120 Hz [Ref], which is adequate for the SHM of a bridge. (Ref: Kim K., Sohn H. Dynamic displacement estimation by fusing LDV and LiDAR measurements via smoothing based Kalman filtering. Mechanical Systems and Signal Processing. 2020.)
  3. Page 4, line 168, “A high-pass Butterworth filter of order three…”A concise explanation is suggested for the filter that you use, specifically, please introduce how the filter removes the low frequency components of the raw data and obtain the filtered displacement.
  4. Page 7, line 213, “Both events can be corrected with more stable UAS control”.According to Figure 7, the measurements of the LDV are easy to be out-of-range due to the hovering motion of the UAS. In practice, there will be strong airflow around the bridge when the trains cross the bridge, and may lead to large motion of the UAS. Therefore, please discuss the effect of the ‘return-to-zero’ process on the displacement accuracy, especially in the condition of large hovering motion.
  5. Page 8, line 230, “At the frequencies greater than 0.5 Hz, the signals from vibrometer as well as the LVDT, show a similar profile.”It is not clear to see the frequency profile that greater than 0.5 Hz in Figure 9. Please give a zoom-in figure for the ‘greater than 0.5 Hz’ component. In addition, please give another figure that compares the filtered displacements in the frequency domain, and shows the natural frequencies of the bridge.
  6. Page 8, line 232, “cut-off frequency of 0.5Hz as explained in Section 2”The cutoff frequency of 0.5Hz is acceptable for railroad timber bridges, which are usually beam bridges. But the first order natural frequency of cable-stayed bridge and suspension bridge are lower than 0.5 Hz, hence the proposed technique is just available for beam bridges.

This study use a timber plant to mock timber bridge, which is not enough to validate the proposed technique in field monitoring. Because the structural characters and the boundary conditions of the plant are quite different from a real bridge. Therefore, it is highly recommended to introduce a test on a real bridge to validate the proposed technique, which should not be too difficult.

Reviewer 2 Report

The paper proposes a UAV-based displacement sensing technique with embedded laser vibrometers for railroad bridge monitoring. The proposed system can be considered as an alternative/extension to existing UAV applications using computer vision for reference-free displacement monitoring, in my opinion. In that sense, I believe that the topic being studied and the proposed scheme has sufficient novelty, besides written well. On the other hand, the verification process seems to be incomplete and missing the essential realistic base, according to the reviewer. I would suggest the following major revisions for consideration of the publication of the manuscript.

1- The introduction is concise yet comprehensive, which is appreciated by the reviewer. One crucial distinction between vision-based UAV and laser vibrometer- based UAV applications seems to be the measurement direction (vertical vs transverse). Is that correct? If so, I would suggest specifying the different throughout the manuscript.

2- Regarding equipment selection, is there any difference between using commonly available quadcopters vs the device utilized in the study (DJI Matrice)? Do they possess the same or similar stability, positioning, and any other aspects? Please discuss.

3- Figure 2 Part b doesn’t seem clear to the reviewer. The figure may need revision.

4- The signal processing issues are a bit too introductory, in my opinion. The application is simply a high-pass filter which may skip some of the presented material (e.g. Figure 4). However, the filter properties (e.g., order) can be reasoned.

5- It didn’t seem clear to me whether the displacement monitoring acts for dynamic identification or engineering demand parameter assessment. In other words, is it essential to extract bridge modal characteristics (if so, may need discussions on how it works with the single-output data)? Or is it a threshold-based application, e.g., a maximum allowed displacement which is being monitored (in this case, the high-pass filter can possess problems since low-frequency displacement would be wiped out)? Please explain.

6- Experimental verification should have been more rigorous, according to the reviewer. The wooden block mimicking the bridge is manually operated as far as I understood. It seems that exploring the limitations and parametric study would suffer from this simplistic scheme, in my opinion. I would suggest a more comprehensive scheme. Otherwise, I would suggest at least a real structure.

7- Similar to the above, the proposed technology addresses a railroad bridge monitoring purpose. However, no such case is presented. I recommend adding a bridge application to complete the validation process.

8- Regarding the results and discussion, I do not recommend mixing SI units with imperial units. Please follow a consistent unit or specify if there are any reasons for combining the two.

9- The statistical findings from three cases are merged in Figure 13. I am not sure if averaging these cases is a viable approach since they possess different conditions. I recommend treating each case differently, so are the statistics.

10- One final comment, which is up to the authors is the naming. Is there a particular reason for choosing UAS rather than UAV? The latter is the common term, therefore unify better with the existing literature. However, the authors can choose the optimal term depending on the need.

In summary, the paper is well narrated, and the topic being studied is new and worthy of investigation. However, the existing material seems incomplete to the reviewer. Validation on a real example is essential, in my opinion. Otherwise, calling out railroad bridge in the title and following may not be appropriate, unless validated in the field. For this reason, I would recommend a revision which could complement the missing aspects.

Author Response

Thank you for your feedback and comments, we have addressed all of them in the attached document
